# Masticatory index for patients wearing dental prosthesis as alternative to conventional masticatory ability measures

**Nareudee Limpuangthip[1], Wacharasak Tumrasvin[1]\*, Chatwarin Sakultae[2]**

**1** Department of Prosthodontics, Faculty of Dentistry, Chulalongkorn University, Bangkok, Thailand, **2** Dental Department, Wiang Pa Pao hospital, Chiang Rai, Thailand

☯ These authors contributed equally to this work.
\* wacharasak.t@chula.ac.th

**Data Availability Statement:** All relevant data are within the paper and its Supporting Information files.

## Abstract

### Objectives

The study objectives were: 1) to develop a masticatory index for evaluating masticatory ability in patients wearing a dental prosthesis, including complete dentures (CD), removable partial denture (RPD), and fixed partial denture (FPD), 2) to test the reliability and validity of the index, and 3) to determine whether the index better reflected patients' masticatory ability compared with conventional subjective and objective measures.

### Methods

The present cross-sectional study consisted of 2 phases: 1) developing the Chulalongkorn University masticatory index (CUMI) consisting of 20 food items in 5 masticatory difficulty grades using a 3-point Likert scale, and 2) application of the CUMI in 110 patients wearing a dental prosthesis, including CD, RPD, and FPD (control group). The CUMI test-retest reliability was reevaluated 2 weeks later. The convergent validity was compared with objective masticatory performance evaluated with a standard peanut chewing test, and subjective eating impacts evaluated by the Oral Impacts on Daily Performances Index. Oral and denture status were determined clinically. The associations between CUMI score, peanut particle size, and eating impact score was identified using Spearman's correlation coefficient. To evaluate discriminant validity, the associations between masticatory ability measurements and oral and denture status were analyzed using regression analyses.

### Results

The CUMI's Cronbach's alpha and intraclass correlation coefficient values were 0.89 and 0.95, respectively. The convergent validity was shown by significant associations between the increased CUMI score, smaller peanut particle size and decreased eating impact score. Multivariable analyses found that the CUMI score, peanut particle size, and percentage of having an eating impact were significantly associated with the number of remaining teeth and posterior occluding pairs, and type and quality of dental prosthesis. However, the CUMI

**Funding:** This research is funded by Chulalongkorn University, Grant number CU_GR_63_11_32_04. Nareudee Limpuangthip is the author who received the funding. The funders had no role in study design, data collection and analysis, decision to publish, or preparation of the manuscript.

**Competing interests:** The authors have declared that no competing interest exist.

demonstrated better discriminant validity because significant dose-response relationships were found only between the decreased CUMI score and increased tooth loss severity, and unacceptable denture quality. Adjusted $R^2$ values of the CUMI models were the highest, followed by those of peanut particle size and eating impact.

## Conclusion

CUMI is a reliable and valid tool to evaluate masticatory ability of patients wearing a dental prosthesis, including CD, RPD, and FPD. Due to a better discriminant validity, the CUMI better reflects masticatory ability of patients compared with conventional subjective and objective masticatory ability measures.

## Introduction

Tooth loss commonly impairs masticatory ability because it alters the types of food selection and limits dietary variety, leading to poor nutritional status [1, 2]. Poor nutrition is a risk for various comorbidity conditions, such as obesity, cardiovascular diseases, and mortality [3–5]. To improve the masticatory ability and quality of life of patients with tooth loss, a dental prosthesis is always the first choice of treatment to replace missing teeth [6, 7]. However, some patients have experienced impaired masticatory ability and quality of life after a period of denture use [6–8]. Therefore, evaluating the masticatory ability of patients wearing dental prosthesis is important for monitoring and maintaining their oral health and quality of life. In this context, a dental prosthesis refers to complete denture (CD), removable partial denture (RPD), and fixed partial denture (FPD).

Several objective and subjective measures have been used to evaluate the masticatory ability of dental patients and those wearing a dental prosthesis [8–10]. The objective measure requires a person to masticate a test food, whereas the subjective measure reflects persons' perception of their ability to eat or chew food [9, 11]. The objective measures include a color-changeable chewing gum [12, 13], number of chewing stroke prior to swallowing [14], and the size of comminuted food particles [8, 10]. However, the objective measures require special equipment and time-consuming to perform [9, 11]. The subjective measures include satisfaction and oral health-related quality of life (OHRQoL) [7–9]. Although the subjective measures require less chair-time and resources, they cannot ensure whether a person can chew a variety of food. Using both subjective and objective measures may better reflect the true masticatory ability of a person than using either of them [9, 11]. However, this may not be practical in routine clinical practice and population-based study. Therefore, it is necessary to develop a single measure to evaluate the masticatory ability of a population to save time, and human and financial resources.

A food questionnaire is one subjective masticatory ability measure; however, it also objectively identifies whether a person can masticate a variety of food. A food questionnaire allows patients to rate the difficulty level in eating food items that vary in hardness and toughness. Several food questionnaire versions have been developed across countries, such as Japan [13, 15], Taiwan [16], Hong Kong [13], and Vietnam [17]. The types and numbers of food items vary depending on the culture and ethnicity. Previously developed food questionnaires have been used specifically with older people [13, 18], partially edentulous people, and complete denture wearers [19–21]. Newly developed questionnaires are commonly validated with objective [13, 19, 20], and subjective masticatory ability measures [13, 15, 21]. However, it has not

been determined whether the newly-developed masticatory index better reflects patients' masticatory ability compared with conventional subjective and objective measures.

The objectives of this study were 1) to develop a masticatory index based on a food questionnaire to evaluate the masticatory ability of patients wearing a dental prosthesis, including CD, RPD, and FPD (control group), 2) to test the reliability and validity of the newly-developed masticatory index using subjective eating impacts and objective masticatory performance as references, and 3) to determine whether the index better reflected patients' masticatory ability compared with conventional subjective and objective measures.

## Materials and methods

### Study design and participants

The present cross-sectional study consisted of two phases. First, a food questionnaire, called the Chulalongkorn University masticatory index (CUMI) was developed. Second, the reliability and validity of the CUMI were determined using objective masticatory performance, and subjective oral impacts on daily performances focusing eating impact as references. The study protocol was approved by the Human Research Ethics Committee of the Faculty of Dentistry (HREC-DCU 2019–081).

The participants were patients who had received conventional prosthodontic treatment from dental students at the Faculty of Dentistry, Chulalongkorn University. The patients had been wearing CD, RPD, or FPD for at least 6 months. A FPD was defined as when at least 26 remaining natural teeth was present. The exclusion criteria were patients who received full mouth rehabilitation (more than 5 units of fixed crowns and/or bridges), had neuromuscular or psychological disorders, or was allergic to peanuts. The participants signed an informed consent prior to participation.

### Phase I: Developing the CU-masticatory index

Initially, 40 patients (37.5% CD, 37.5% RPD, and 10% FPD and dentate) were asked about the typical food types they had eaten during the past week, and the food types they had difficulty in chewing or would like to eat but could not chew at all. Then, the additional 20 patients were interviewed. We found that the most regularly-consumed and rarely-consumed foods obtained from 40 and 60 patients were similar. Therefore, the Phase I comprised a total of 60 patients (40% CD, 40% RPD, 10% FPD and 10% dentate individuals) with mean age (±s.d.) = 67.7 ±8.6 years. A higher proportion of removable dentures was included because they reported a greater variety of food types and textures compared with the FPD and dentate individuals. A greater variety of food type was due to the variation in denture quality and oral status among the RPD and CD wearers. Then, 80 food types were obtained from the interview.

From all food types, the 14 most frequently-consumed food items covering 4 food groups were selected and included in the questionnaire as follows:

1. Protein-rich foods: minced pork, boiled egg, omelet, fried chicken, and crispy pork

2. Carbohydrate-rich foods: steamed rice, noodles, porridge, and sticky rice

3. Vegetables: boiled cabbage and stir-fried kale

4. Fruits: orange, banana, and guava

The 6 most common foods that the patients had difficulty in chewing or would like to eat but could not chew at all were: stir-fried morning glory, dried shrimp, stir-fried water mimosa, rice cake, kalamare (Thai caramel-like toffee), and grains or seeds, such as sesame seed, ground

peanut, and roasted rice powder. Therefore, the CUMI comprised 20 food types covering both frequently-consumed and rarely-consumed foods.

## Phase II: CUMI Application

One-hundred and ten participants who did not participate phase I enrolled in phase II. A test-retest reliability of the questionnaire was evaluated by re-interviewing 20 participants to determine their masticatory score on two occasions with a 2-week interval. The questionnaire validity was tested using both subjective and objective measures of masticatory ability, which were eating impact and a standard peanut mastication, respectively, as a reference.

**CUMI assessment.** The participants rated the level of difficulty in eating/chewing the 20-food items using a 3-point Likert scale: can chew well (2), can chew with difficulty (1), and cannot chew at all (0). Any food item that the patient had never eaten or could not remember, was recorded as a missing item and was not included in the score calculation. The masticatory difficulty score of each food item, ranging from 0–2, was determined from the average score obtained from all participants. Ranging from the highest to the lowest level of masticatory difficulty score, 20 food items were categorized into 5 masticatory difficulty grades with 4 food items in each grade. The masticatory difficulty score of each grade was calculated from the average score of 4 food items in that grade. Because the masticatory difficulty ratio of grade I was l.00, the masticatory difficulty ratio of the other grades was calculated as the masticatory difficulty score of grade I divided by that grade. The CU-masticatory score of each participant was calculated using the following formula [20]:

$$\text{CUMI score (\%)}$$
$$= \frac{\sum(\text{Masticatory difficulty ratio of a food grade x Masticatory score of the food grade of each person})}{(\text{Total masticatory difficulty ratio of all food grades x2})} \text{x100\%}$$

The food item that a patient had never eaten or could not remember was not included in the score calculation.

In addition to the participants' responses, three experts in prosthodontics scored the masticatory difficulty grade of each food item, and the values were compared to those obtained from all participants. Weighted Kappa scores ranging from 0.75–0.87 were calculated, indicating 90–95% agreement between the experts and participants.

**Subjective masticatory ability: OHRQoL assessment.** The OHRQoL was assessed by a face-to-face interview using the Thai version of oral impacts on daily performances (Thai-OIDP) which has been validated in a Thai population [22, 23]. The measurement focuses on oral conditions that affect the ability to carry out eight daily activities within three performances: physical (eating, speaking/pronouncing clearly, cleaning teeth/denture/oral cavity), psychological (sleeping/relaxing, smiling/laughing/showing teeth without embarrassment, maintaining usual emotion), and social (performing work, and contacting people). The participants rated the frequency and severity of the impact. The participant was classified as had no oral impact (OIDP score = 0) or had an oral impact (OIDP score > 0), as well as no eating impact (eating impact score = 0) and had an eating impact (eating impact score > 0).

**Objective masticatory ability: Masticatory performance assessment.** Masticatory performance was assessed using a multiple sieve method of peanut mastication [24, 25]. The patients sat in an upright position and masticated 3 g of roasted peanuts for 20 strokes in triplicate. The comminuted peanut particles were sieved using 12 standard test sieves that were placed on a vibrating sieve shaker at a frequency of 70 Hz for 3 min. The peanut particles that did not pass through the test sieves were collected and calculated to determine the median peanut particle size. The median peanut particle size was defined as the sieve diameter through

which 50% of the comminuted particles passed: the smaller median particle size, the better masticatory performance.

**Covariate assessment.** The oral and denture status of the patients were examined. The oral status comprised the number of remaining natural teeth (less than 20, or at least 20 teeth), posterior occluding pairs (less than 4, or at least 4 occluding pairs), and edentulous condition (dentate, partial edentulism, and complete edentulism). The participants were categorized as dentate when at least 26 natural teeth remained.

The type of dental prosthesis was categorized into 3 types; CD, RPD and FPD. When more than one type of dental prosthesis was present, it was categorized as the type with the greater severity of tooth loss. The FPD group, dentate individuals who had at least 26 remaining natural teeth, was served as a positive control. The clinical quality of the removable dentures based on retention and stability was examined by one calibrated prosthodontist. Retention and stability were evaluated because an ill-fitting denture is the most common problem for removable denture wearers [26, 27]. Retention and stability of CDs were evaluated according to the CU-modified Kapur criteria [24], while those of the RPDs were evaluated based on criteria modified from the CU-modified Kapur criteria and NHANES III (S1 and S2 Tables) [26]. Retention and stability levels were scored using 4-point and 3-point Likert scales, respectively. The maxillary and mandibular denture quality was categorized as acceptable or unacceptable. The clinical quality of the denture was considered as acceptable when the retention and stability of both maxillary and mandibular denture were acceptable. If either or both dentures were unacceptable, the overall denture quality was considered unacceptable [24]. The intra-examiner reliability in denture quality evaluation was examined in 20 denture wearers with a 2-week interval. The Kappa score ranged from 0.90–0.95, which indicated excellent intra-examiner reliability.

**Power analysis.** The study power was calculated using G*Power version 3.1.9.2 (Heinrich-Heine-Universität Düsseldorf, Düsseldorf, Germany) based on the hypothesis that the CUMI score would be significantly different between the three types of dental prosthesis. Our results indicated that the CUMI score (mean ±sd.) of the participants wearing FPD ($n_1 = 19$), RPD ($n_2 = 56$) and CD ($n_3 = 35$) were 98.1% (±4.5), 82.2% (±12.3) and 63.7% (±17.0), respectively. Using the F test for analysis of variance (ANOVA), a 99.9% power was calculated at $\alpha = 0.05$.

## Statistical analysis

The data were analyzed using STATA version 13.0 (StataCorp LP, College Station, TX, USA) at a 5% significance level. Descriptive analyses were calculated as mean (±sd.) and percentage distribution (%). The internal consistency of the CUMI was analyzed using Cronbach's alpha, and the test-retest reliability was analyzed using an intraclass correlation coefficient (ICC). The convergent validity of the CUMI was analyzed with Pearson's correlation coefficient using peanut particle size and an eating impact score as references. The discriminant validity of the CUMI score, peanut particle size, and oral impact was determined by evaluating their associations with oral- and denture-related variables using bivariate and multivariable analyses. For bivariate analysis, differences in the CUMI score and peanut particle size between each variable were determined using one-way ANOVA and Tukey's post hoc comparison test, whereas differences in the percentage of overall and eating impacts was determined using the Chi-square or Fisher's exact test. Adjusting for covariates, the association between oral and denture status, and the CUMI score, peanut particle size, and having an eating impact were analyzed using multiple linear and logistic regression. In addition, the percentage of food items that the participants reported difficulty or inability to chew was determined between different conditions of eating impacts, as well as types and qualities of the dental prosthesis.

## Results

The developed CUMI consisted of 20 food items in 5 masticatory difficulty grades, ranging from grade I (most easily chewed) to grade V (most difficult to chew) (Table 1). The participants attending phase II had a mean age of 65.0 ±8.9 years (range 37–85 years). Approximately 32%, 51% and 17% of the participants wore CD, RPD, and FPD, respectively. Calculated from the average masticatory difficulty score of all participants, the masticatory difficulty ratio of each grade was obtained. The CUMI score of each participant was determined using the formula: CUMI score (%) = (a + 1.02b + 1.21c + 1.45d + 1.61e)/12.58 ×100%; when the average masticatory difficulty score of the food grade I–IV of each person were a, b, c, d, and e, respectively. The food item that a patient had never eaten or could not remember was not calculated: the higher CUMI score, the higher masticatory ability.

The internal consistency of the CUMI based on the Cronbach's alpha value was 0.89. The test-retest reliability of the CUMI based on ICC value was 0.95 (95% CI = 0.88–0.98). The convergent validity based on the Pearson's correlation coefficient (r) revealed statistically significant correlations between a higher CUMI score and smaller peanut particle size, and lower eating impact score (r = -0.66 and -0.57, respectively (p < 0.001)).

The discriminant validity of the CUMI score, peanut particle size, and eating impacts was determined by comparing the outcomes of the participants with different oral and denture status. Univariate analyses revealed that decreased CUMI score and increased peanut particle size were found in the older age groups (Table 2). The CUMI score, peanut

**Table 1. CUMI evaluation.**

| Masticatory difficulty grade | Food items | Masticatory difficulty score of each food item: mean (±sd.) | Masticatory difficulty score of each food grade: mean (±sd.) | Masticatory difficulty ratio | Average masticatory difficulty point[†] |
|---|---|---|---|---|---|
| I | Porridge | 1.99 (±0.10) | 1.98 (±0.12) | 1.00 | a |
| | Omelet | 1.98 (±0.13) | | | |
| | Boiled cabbage | 1.98 (±0.13) | | | |
| | Banana | 1.98 (±0.13) | | | |
| II | Steamed rice | 1.95 (±0.21) | 1.95 (±0.22) | 1.02 | b |
| | Boiled egg | 1.95 (±0.21) | | | |
| | Noodle | 1.94 (±0.23) | | | |
| | Minced pork | 1.94 (±0.23) | | | |
| III | Orange | 1.80 (±0.42) | 1.63 (±0.53) | 1.21 | c |
| | Fried chicken | 1.66 (±0.53) | | | |
| | Sticky rice | 1.60 (±0.56) | | | |
| | Stir-fried morning glory | 1.46 (±0.62) | | | |
| IV | Stir-fried kale | 1.41 (±0.67) | 1.37 (±0.67) | 1.45 | d |
| | Grains or seeds | 1.37 (±0.68) | | | |
| | Dried shrimp | 1.36 (±0.65) | | | |
| | Guava | 1.34 (±0.70) | | | |
| V | Crispy pork | 1.31 (±0.62) | 1.23 (±0.68) | 1.61 | e |
| | Rice cake | 1.31 (±0.67) | | | |
| | Stir-fried water mimosa | 1.17 (±0.70) | | | |
| | Kalamare | 1.14 (±0.74) | | | |

[†]Average masticatory difficulty point was calculated from average score of 4 food items in that grade, excluding the food item which had never been eaten or could not be remembered.

**Table 2. Masticatory ability of the participants.**

| Variables | Distribution | CUMI score (%): | Median peanut particle size (mm): | Having oral impact (%): | |
|---|---|---|---|---|---|
| | (%) | mean (±sd.) | mean (±sd.) | Overall oral impact | Eating impact |
| *All participants* | | *79.1 (±17.7)* | *2.3 (±0.9)* | *49.1* | *45.5* |
| **Age (years):** < 60 | 27.3 | 88.9 (±11.3)** | 1.8 (±0.6)* | 24.1 | 20.0 |
| 60–69 | 35.4 | 80.2 (±17.7) | 2.1 (±0.7) | 27.8 | 28.0 |
| > 69 | 37.3 | 70.8 (±17.8) | 2.8 (±1.1) | 48.1 | 52.0 |
| **Sex:** Male | 40.0 | 76.4 (±18.8) | 2.4 (±1.1) | 40.7 | 44.0 |
| Female | 60.0 | 80,8 (±16.8) | 2.2 (±0.8) | 59.3 | 56.0 |
| **Oral status:** | | | | | |
| • Number of remaining teeth and occluding pairs: | | | | | |
| • ≥ 20 teeth and ≥ 4 occluding pairs | 39.1 | 93.0 (±8.9)** | 1.8 (±0.6)* | 22.2* | 20.0* |
| • ≥ 20 teeth and < 4 occluding pairs | 7.3 | 82.3 (±12.1) | 2.1 (±0.5) | 5.6 | 6.0 |
| • < 20 teeth | 53.6 | 68.5 (±16.0) | 2.7 (±1.1) | 72.2 | 74.0 |
| • Edentulous condition: | | | | | |
| • Dentate | 17.3 | 98.1 (±4.5)** | 1.6 (±0.3)* | 7.4* | 6.0* |
| • Partial edentulism | 50.9 | 82.2 (±12.3) | 2.1 (±0.6) | 48.2 | 46.0 |
| • Complete edentulism | 31.8 | 63.7 (±17.0) | 3.1 (±1.2) | 44.4 | 48.0 |
| **Type and quality of dental prosthesis:** | | | | | |
| • Fixed partial denture | 17.3 | 98.1 (±4.5)** | 1.6 (±0.3)* | 7.4* | 6.0* |
| • Removable partial denture: acceptable quality | 31.8 | 85.0 (±11.4) | 2.0 (±0.5) | 20.3 | 16.0 |
| unacceptable quality | 19.1 | 77.5 (±12.5) | 2.3 (±0.7) | 27.8 | 30.0 |
| • Complete denture: acceptable quality | 15.4 | 68.6 (±16.4) | 2.6 (±1.0) | 13.0 | 14.0 |
| unacceptable quality | 16.4 | 59.0 (±16.6) | 3.5 (±1.2) | 31.5 | 34.0 |

*Significant difference at **$p < 0.001$, *$p < 0.05$.

particle size and percentage of having eating impacts were significantly different between oral and denture status.

The multiple regression analyses of each masticatory ability measure were split into two models because there was a collinearity between tooth loss status and type of dental prosthesis (Table 3). After adjusting for age and sex, there was a significant dose-response relationship between an increased CUMI score, greater tooth loss severity, and unacceptable denture quality. Meanwhile, a dose-response relationship was not shown in the peanut particle size and eating impact models. For type of dental prosthesis model, the peanut particle size was significantly different only between FPD and CD, whereas the oral impact was different only between the acceptable and unacceptable denture quality. For both oral and denture status models, the adjusted $R^2$ values of the CUMI outcome was the highest, followed by those of peanut particle size and eating impacts. Therefore, the CUMI demonstrated better discriminant validity than the peanut particle size and eating impact models.

Among the participants who had no eating impact, approximately 30–50% of them reported difficulty or inability to chew food items in grade IV and grade V, and stir-fried morning glory in grade III (Table 4). The participants with an eating impact were more likely to report difficulty or inability to chew food items in grade III–V, compared with those without an eating impact. Difficulty or inability to chew food items in grade III–V was most frequently reported in participants wearing CDs, followed by RPDs and FPDs, and more frequently reported in participants with an unacceptable denture quality compared with those wearing an acceptable quality denture.

**Table 3. Multivariable analyses of masticatory ability measures and relating variables.**

| Variables | CUMI score (adjusted β) | | Peanut particle size (mm) (adjusted β) | | Having eating impact (adjusted OR) | |
|---|---|---|---|---|---|---|
| | Model 1 | Model 2 | Model 1 | Model 2 | Model 1 | Model 2 |
| **Number of remaining teeth and occluding pairs:** | | | | | | |
| • ≥ 20 teeth and ≥ 4 occluding pairs | 0 (ref) | | 0 (ref) | | (ref) | |
| • ≥ 20 teeth and < 4 occluding pairs | -11.6 (-21.7, -1.4)* | - | 0.4 (-0.3, 1.0) | - | • (0.1, 2.0) | - |
| • < 20 teeth | -21.8 (-27.7, -15.9)* | | 0.6 (0.3, 1.0)* | | 0.2 (0.1, 0.6)* | |
| **Type and quality of dental prosthesis:** | | | | | | |
| • Fixed partial denture | | 0 (ref) | | 0 (ref) | | (ref) |
| • Removable partial denture: | | | | | | |
| • acceptable quality | | -11.9 (-19.3, -4.4)* | | 0.3 (-0.2, 0.7) | | • (0.4, 8.0) |
| • unacceptable quality | - | -17.6 (-26.9, -8.3)* | - | 0.4 (-0.1, 1.0) | - | 15.7 (2.5, 100.0)* |
| • Complete denture: | | | | | | |
| • acceptable quality | | -27.1 (-36.7, -17.4)* | | 0.8 (0.2, 1.4)* | | • (0.7, 32.0) |
| • unacceptable quality | | -37.1 (-46.4, -27.7)* | | 1.7 (1.1, 2.3)* | | 116.3 (55.9, 152.0)* |
| **Adjusted $R^2$** | 43.7% | 48.1% | 23.4% | 37.3% | 12.6% | 28.3% |

## Discussion

This study developed a single masticatory index, called the CUMI, to evaluate the masticatory ability of patients with different oral and denture status. The internal consistency and test-retest reliability were identified. A convergent validity was verified as reference to the conventional subjective eating impact and objective masticatory performance. Since significant dose-

**Table 4. Food items with chewing difficulty or could not be chewed according to different status of eating impact, type and quality of dental prosthesis.**

| Food items with chewing difficulty or could not be chewed | Eating impact | | Type and quality of dental prosthesis | | | | |
|---|---|---|---|---|---|---|---|
| | Presence | Absence | CD | | RPD | | FPD |
| | (n = 50) | (n = 60) | Unacceptable (n = 18) | Acceptable (n = 17) | Unacceptable (n = 21) | Acceptable (n = 35) | (n = 19) |
| Grade I: Porridge | 2.1 | 0.0 | 5.6 | 0.0 | 0.0 | 0.0 | 0.0 |
| Omelet | 4.0 | 0.0 | 11.1 | 0.0 | 0.0 | 0.0 | 0.0 |
| Boiled cabbage | 0.0 | 0.0 | 11.1 | 0.0 | 0.0 | 0.0 | 0.0 |
| Banana | 4.0 | 0.0 | 11.1 | 0.0 | 0.0 | 0.0 | 0.0 |
| Grade II: Steamed rice | 10.0 | 0.0 | 27.8 | 0.0 | 0.0 | 0.0 | 0.0 |
| Boiled egg | 4.0 | 3.3 | 16.7 | 5.9 | 4.8 | 0.0 | 0.0 |
| Noodle | 8.2 | 3.3 | 11.1 | 6.2 | 14.3 | 0.0 | 0.0 |
| Minced pork | 10.0 | 1.7 | 11.1 | 11.8 | 4.8 | 2.9 | 0.0 |
| Grade III: Orange | 36.0 | 5.0 | 44.4 | 23.5 | 38.1 | 0.0 | 0.0 |
| Fried chicken | 52.1 | 15.0 | 77.8 | 52.9 | 36.8 | 11.4 | 0.0 |
| Sticky rice | 53.1 | 22.4 | 66.7 | 56.2 | 42.1 | 28.6 | 0.0 |
| Stir-fried morning glory | 64.0 | 33.3 | 77.8 | 76.5 | 47.6 | 40.0 | 5.3 |
| Grade IV: Stir-fried kale | 69.4 | 31.7 | 94.1 | 64.7 | 57.1 | 34.3 | 10.5 |
| Grains or seeds | 74.0 | 31.6 | 88.9 | 82.3 | 65.0 | 36.4 | 16.7 |
| Dried shrimp | 76.6 | 35.7 | 93.7 | 78.6 | 75.0 | 41.2 | 5.3 |
| Guava | 77.6 | 32.2 | 94.4 | 75.0 | 57.1 | 47.1 | 0.0 |
| Grade V: Crispy pork | 79.2 | 44.8 | 88.9 | 81.2 | 73.7 | 51.4 | 16.7 |
| Rice cake | 80.4 | 38.6 | 88.2 | 82.3 | 73.7 | 46.9 | 5.6 |
| Stir-fried water mimosa | 83.0 | 51.7 | 100.0 | 100.0 | 68.4 | 62.9 | 10.5 |
| Kalamare | 79.6 | 51.1 | 93.3 | 87.5 | 72.2 | 64.0 | 11.8 |

response relationships were found only between an increased CUMI score and greater tooth loss severity, and unacceptable denture quality, the CUMI demonstrated better discriminant validity than the conventional subjective and objective measures. The results indicated that the developed masticatory index better reflected patients' masticatory ability compared with conventional subjective and objective measures.

The food items in previously developed questionnaires were commonly selected from regularly consumed food [16], or shorten from multiple food items or an original food book [13, 15]. The food items were selected by a focus group of dentists [17], or together with patient participation [28]. However, the CUMI comprises both regularly- and rarely-consumed food reported by the patients alone. The inclusion of food items with different chewing difficulty levels was to improve the discriminant validity in differentiating severities of tooth loss and denture status. The different masticatory difficulty grades were verified between dental experts and patients. However, we did not use any specific instrument to assess masticatory difficulty level of each food item because it includes mixed properties of the food such as hardness, toughness, stickiness, slipperiness, and fibrousness. Therefore, no specific instrument can comprehensively determine these properties and verify the outcome. Similar to a previous food questionnaire in CD wearers [20], the CUMI score was calculated from a weighted score of each food grade; the more toughness and hardness, the greater values were weighed. The food items that the patients had never eaten or could not remember eating were not included in the score calculation to reduce bias from patients' preference. In addition, the food items cover the four basic macronutrients for further use in evaluating nutritional status.

The convergent validity of the CUMI was evaluated using both subjective eating impact and objective masticatory performance as references, and the moderate correlations between the CUMI score and the referent measures were found. The results indicate that the CUMI could be used for evaluating masticatory ability in patients wearing a dental prosthesis in comparison with the conventional subjective and objective measures. The OHRQoL was used as subjective outcome because it is a cross-cultural validated tool, thus, the findings can be generalized to other populations [29]. Our results demonstrated a stronger association between the CUMI and eating impact score (r = -0.57), compared with that of previously developed food questionnaires and the Oral Health Impacts Profile-14 (OHIP-14, r = -0.46) [18], and the Geriatric Oral Health Assessment Index (GOHAI, r = 0.48) [15]. This difference might be because the present study focused on an eating impact rather than examining the overall OHRQoL, and the OIDP focuses on the ultimate impact, rather than pain and discomfort, that might not affect chewing ability [22]. Although multiple sieve method of peanut mastication is worldwide used, it consumes more time and resources to perform than the subjective measures. The whole evaluation process takes 2 days to obtain the result since collecting the comminuted peanut particles from patients, drying the comminuted peanut particles overnight, and then, analyzing the peanut particle size on the next day. Therefore, peanut mastication might be practical only in clinical study or research.

From the multivariable analysis models, the discriminant validity of the CUMI in identifying different oral and denture status was better than those of eating impacts and masticatory performance. The explanation is that there was a significant dose-response relationship between increased CUMI score and decreased tooth loss severity, and acceptable denture quality. Although a dose-response relationship was also found in the masticatory performance models, this was not significant. Additionally, the adjusted $R^2$ values of the CUMI models were the highest, followed by those of the masticatory performance and eating impacts. The values indicated that oral and denture status better explained the variances in the CUMI score compared with masticatory performance and eating impacts. A previous study in CD wearers consistently found that an objective masticatory performance better reflected patients'

masticatory ability compared with eating impacts [8]. These results might be because the masticatory performance evaluation using a single food may not reflect the ability to chew a variety of food in daily life [11, 20]. In addition to oral and denture status, masticatory performance is affected by an individual's bite force and masticatory muscle thickness [8, 30]. Furthermore, we found that up to 70–80% of the participants without eating impact could not or chew the food items in grade IV to V. Misinterpretation of eating impacts may occur when a person adapts to a soft diet without perceiving any eating or chewing problems. Based on the above reasons, the CUMI may better reflect the masticatory ability of patients wearing a dental prosthesis compared with the conventional subjective oral impacts and objective masticatory performance measures.

The CUMI may assist in denture quality evaluation without requiring a dental professional or trained personnel to evaluate denture retention and stability. In this study, the FPD group served as a positive control because they showed the least frequent eating impact, and the highest masticatory performance and CUMI score. Approximately 79% of them could easily chew all food items or get a full CUMI score. Difficulty or inability to chew food items in grade I or II indicates increased severity of tooth loss and denture quality compared with the inability to chew those in grade III–V. Difficulty or inability in chewing food items in grade III–V, and those in grade III and IV was more likely to be reported by patients with an unacceptable RPD quality and unacceptable CD quality, respectively. It was noted that most CD wearers had difficulty or were unable to chew grade V food items and grains or seeds regardless of denture quality. Therefore, to maximize the masticatory ability of CD wearers, mandibular two-implant overdentures should be recommended as the first-choice standard of care for edentulous patients [31].

The food items in the present questionnaire are mostly Asian food, however, it is considered for worldwide use because the Asian-living culture and Asian populations are prevalent worldwide. In addition, the present study aimed not only to develop the questionnaire, but also to propose a concept of developing a questionnaire for masticatory ability evaluation in patients wearing different types and qualities of dental prosthesis. Although the types and number of food items may vary among cultures and environments, the questionnaire should comprise both regularly- and rarely-consumed foods. Food textures should be varied in hardness and toughness, and include sticky, grainy, fibrous foods. Validity testing should be performed using both subjective OHRQoL and objective masticatory ability as references. The discriminant ability of the masticatory index in identifying different tooth loss severities and denture qualities helps determine whether a developed food questionnaire better reflects masticatory ability compared with the conventional subjective and objective measures. Therefore, the CUMI can be used to evaluate the masticatory ability of patients with different oral status and dental prosthesis worn, both in clinical practice and population-based studies. It may be used as a screening tool for determining the priority of prosthodontic treatment need, and dietary consultation.

Some limitations of this study were noted. We did not identify the responsiveness of the CUMI by evaluating if it changed after prosthodontic treatment. Despite including food items with a variety of nutritional types, the association between the CUMI and nutritional status was not investigated. Although this study demonstrated 99% power of sample size, the number of participants might be too small to identify whether the CUMI could differentiate subgroups of some independent variables, such as dental status, and severity of partial and complete edentulous conditions. Further studies should determine the responsiveness of the CUMI before and after prosthodontic treatment. Sample size should be increased to improve the generalizability of the findings in order to utilize the CUMI in various groups of patients with

more oral and physical complexity, such as implant-retained overdentures and disabled patients.

## Conclusions

Within the limitation of this study, particularly the number and variety of participants, the CUMI can be used as a valid and reliable masticatory index to differentiate different types of dental prostheses and removable denture qualities based on retention and stability. It better reflects masticatory ability in patients wearing a dental prosthesis compared with subjective eating impacts and objective masticatory performance based on peanut mastication.

## Supporting information

**S1 Table. Criteria for evaluating the retention and stability of removable partial dentures (RPD) as modified from the CU-modified Kapur index and NHANES III.** (DOCX)

**S2 Table. Criteria for evaluating the clinical quality of removable complete and partial dentures (modified from CU-modified Kapur).** (DOCX)

**S1 Data.** (XLSX)

## Acknowledgments

The authors gratefully acknowledge Dr. Kevin Tompkins for language revision of the manuscript.

## Author Contributions

**Conceptualization:** Nareudee Limpuangthip, Wacharasak Tumrasvin.

**Data curation:** Nareudee Limpuangthip, Chatwarin Sakultae.

**Formal analysis:** Nareudee Limpuangthip, Chatwarin Sakultae.

**Funding acquisition:** Nareudee Limpuangthip.

**Investigation:** Wacharasak Tumrasvin, Chatwarin Sakultae.

**Methodology:** Nareudee Limpuangthip, Wacharasak Tumrasvin, Chatwarin Sakultae.

**Project administration:** Wacharasak Tumrasvin.

**Resources:** Nareudee Limpuangthip, Wacharasak Tumrasvin.

**Software:** Nareudee Limpuangthip.

**Supervision:** Nareudee Limpuangthip, Wacharasak Tumrasvin.

**Validation:** Nareudee Limpuangthip, Wacharasak Tumrasvin, Chatwarin Sakultae.

**Visualization:** Nareudee Limpuangthip, Wacharasak Tumrasvin, Chatwarin Sakultae.

**Writing – original draft:** Nareudee Limpuangthip, Wacharasak Tumrasvin, Chatwarin Sakultae.

**Writing – review & editing:** Nareudee Limpuangthip, Wacharasak Tumrasvin, Chatwarin Sakultae.

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
