## [Decision Letter · Decision Letter 0]

24 Feb 2021

PONE-D-21-02795

Masticatory Index for Patients Wearing Dental Prosthesis as Alternative to Conventional Masticatory Ability Measures

PLOS ONE

Dear Dr. tumrasvin,

Thank you for submitting your manuscript to PLOS ONE. After careful consideration, we feel that it has merit but does not fully meet PLOS ONE’s publication criteria as it currently stands. Therefore, we invite you to submit a revised version of the manuscript that addresses the points raised during the review process.

please address the comments on the low report rate of food, the power calculation, and the food global representation of this study.

We look forward to receiving your revised manuscript.

Kind regards,

Sompop Bencharit, DDS, MS, PhD, FACP

Academic Editor

PLOS ONE

Journal Requirements:

Reviewers' comments:

Reviewer's Responses to Questions

**Comments to the Author**

1. Is the manuscript technically sound, and do the data support the conclusions?

Reviewer #1: Partly

2. Has the statistical analysis been performed appropriately and rigorously? 

Reviewer #1: No

3. Have the authors made all data underlying the findings in their manuscript fully available?

Reviewer #1: No

4. Is the manuscript presented in an intelligible fashion and written in standard English?

Reviewer #1: Yes

5. Review Comments to the Author

Reviewer #1: Though the topic is of general interest I really do not see what new aspects relevant to the international scientific community are presented.

The authors developed a food questionnaire based to the reports of only sixty individuals where 40 % were provided with complete dentures and RPDs and only 10% FPD or dentate. This may be already decisively biased as it is known that especially patients with CDs tend to select food that they can mince.

I am not a biostatistician but I doubt the power calculation. I have never seen a power calculation with a power of 0,99 and alpha = 0.05 on the basis of only 110 subjects. Even more as the Stds are quite high.

As the food items are special at least for Asian food the approach is very limited on worldwide perspective.

On the other hand, peanuts that the authors use as a comparison are more or less available worldwide.

The different chewing difficulty levels claimed for the food items were never assessed and /or verified.

With regard to the results I really do not see when compared to the peanuts test

The statement that the developed masticatory index reflects the patients’ ability better the conventional subjective or objective methods is not supported by the data.

6. PLOS authors have the option to publish the peer review history of their article (what does this mean?). If published, this will include your full peer review and any attached files.

Reviewer #1: No

---

## [Author Response · Author response to Decision Letter 0]

1 Mar 2021

Response to reviewer

The authors are pleased to submit our revised manuscript ID. PONE-D-21-02795, entitle ‘Masticatory Index for Patients Wearing Dental Prosthesis as Alternative to Conventional Masticatory Ability Measures’. The requested revisions have been made in the manuscript in track changes, and our point-by-point responses are below.:

1. Comment: Though the topic is of general interest I really do not see what new aspects relevant to the international scientific community are presented.

Response: The present study demonstrated the concept of developing a single masticatory index using a food questionnaire for evaluating masticatory ability of patients with various oral and denture status. Several new aspects have been proposed as follows: 

1) Food questionnaires developed in previous studies included only the regularly-consumed foods. In contrast, the present study suggested developing the index that comprises both regularly- and rarely-consumed foods due to chewing difficulty. In addition, the food types should vary in hardness and toughness, and include sticky, grainy, fibrous foods. This is to improve the ability of the masticatory index to discriminate the patients with different oral and dental prosthesis status, in other words, to improve the ‘discriminant validity’ of the index. The description has been written in the ‘Discussion’ section (Page 15, 1st paragraph)

Note: Discriminant validity measures constructs that theoretically should not be highly related or correlated to each other. The measurement with high discriminant validity would be able to discriminate the variables or constructs that are not correlated to each other.

 2) In previous studies, masticatory index was validated with either a conventional subjective or objective masticatory ability measure by determining their associations with the conventional measures, in other words, determining the ‘convergent validity’. The convergent validity indicates that the index is as appropriate as the conventional masticatory ability measure.

In contrast, the present study determined not only the ‘convergent validity’, but also the ‘discriminant validity’ of the index compared with the conventional subjective and objective measures using the multiple regression analyses (Table 3). The discriminant ability of the masticatory index in identifying different tooth loss severities and denture qualities helps determine whether a developed food questionnaire better reflects masticatory ability compared with the conventional subjective and objective measures. The description has been written in the ‘Discussion’ section (Page 16, 1st paragraph).

Therefore, despite the food items may vary among cultures, the concept of questionnaire development in this study can be applied worldwide for other studies with different cultures or environments to create their own masticatory indicator to be used in routine clinical practice and population-based study. The description has been written in the ‘Discussion’ section (Page 17, 2nd paragraph)

2. Comment: The authors developed a food questionnaire based to the reports of only sixty individuals where 40 % were provided with complete dentures and RPDs and only 10% FPD or dentate. This may be already decisively biased as it is known that especially patients with CDs tend to select food that they can mince. 

Response: 

2.1) Our pilot study at Phase I included 40 patients (15 CD, 15 RPD, 10 FPD and dentate). They were interviewed for the regularly-consumed and rarely-consumed foods. Then, the additional 20 patients (total = 60) were also interviewed. We found that the most regularly-consumed and rarely-consumed foods obtained from 40 and 60 patients were similar. 

Therefore, the reason for using only 60 individuals because the food items would be similar to those obtained from more than 60 patients.

2.2) The CUMI comprised both regularly-consumed and rarely-consumed foods. The inclusion of only 10% FPD and 10% dentate was due to the following reasons:

- The inclusion of more removable dentures was because they reported a greater variety of food types and textures compared with FPD and dentate individuals. The wide variety of food type was due to the variation in denture quality and oral status among the RPD and CD wearers. The increased food varieties help improve discriminant validity of the masticatory index.

 - This questionnaire consists of both regularly-consumed and rarely-consumed foods due to chewing difficulty. In addition, not only CD but also RPD, FPD and dentate were interviewed. Therefore, we ensure that a selective bias reported by patients was minimized.

3. Comment: I am not a biostatistician but I doubt the power calculation. I have never seen a power calculation with a power of 0,99 and alpha = 0.05 on the basis of only 110 subjects. Even more as the Stds are quite high.

Response: The F test for analysis of variance (ANOVA) was used to determine the effect of different means between three groups. A type of power analysis was a post hoc comparison test to compute the achieved power, given an alpha, sample size and effect size. 

To calculate the effect size, the mean CUMI values of the three groups (CD = 98.1, RPD = 82.2, and FPD = 63.7) and the highest standard deviation (s.d.) value among three groups (17.0) were used. Then, an effect size of 0.7027 was calculated. Given that the effect size = 0.7027, an alpha = 0.05, and a total sample of 110 within 3 groups, the power of 99% was achieved. The relatively high power is due to the significant differences between the FPD and the CD group. The calculation was shown in the below figure. 

4. Comment: As the food items are special at least for Asian food the approach is very limited on worldwide perspective. On the other hand, peanuts that the authors use as a comparison are more or less available worldwide.

Response: The food items in the present questionnaire are special for Asian food, however, it is considered for worldwide use because the Asian-living culture and Asian populations are prevalent worldwide. In addition, the present study aimed not only to develop the questionnaire, but also to propose a concept of developing a questionnaire for masticatory ability evaluation in patients wearing different types of dental prosthesis. The authors believe that the readers can apply the concept and protocol of the present study to develop their own questionnaires, to be used in other cultures and countries.

 Although peanut mastication is worldwide used, it is not practical in routine dental practice and population-based study because it requires special equipment and time-consuming to perform (The description has been written in the ‘Introduction’ section, Page 4, 1st Paragraph). The special equipment includes standard test sieves and vibrating sieve shaker. Peanut test is time consumption since it takes 2 days for evaluation process, from obtaining peanut particles comminuted by patients, drying peanut particles for 24 hours, and then analyzing the peanut particle size on the next day to obtain the result. Additional personal is also required to perform the evaluation process. Therefore, peanut mastication is practical only in clinical study or research, while the developed CUMI is more practical to be used in routine clinical practice and population-based study.

5. Comment: The different chewing difficulty levels claimed for the food items were never assessed and /or verified.

Response: The different chewing difficulty levels have never been assessed/verified by any special instruments or equipment. This was because chewing difficulty level is the outcome which includes mixed properties of the food, for example, hardness, toughness, stickiness, slipperiness, fibrousness, and so on. Therefore, no specific instrument can be used to verify the different chewing difficulty levels of the food items. 

However, the index was verified by three experts in prosthodontics, and determined their agreement in grading levels of chewing difficulty for each food item. 

6. Comment: With regard to the results, I really do not see when compared to the peanuts test. The statement that the developed masticatory index reflects the patients’ ability better the conventional subjective or objective methods is not supported by the data.

Response: The comparison between the CUMI, peanut test, and eating impact has been performed by using the multiple regression analyses to determine the ‘discriminant validity’ of each measure (Table 3). The results showed that after adjusting for age and sex, there was a significant dose-response relationship between an increased CUMI score, greater tooth loss severity, and unacceptable denture quality. Meanwhile, a dose-response relationship was not shown in the peanut particle size and eating impact models. For type of dental prosthesis model, the peanut particle size was significantly different only between FPD and CD, whereas the oral impact was different only between the acceptable and unacceptable denture quality. For both oral and denture status models, the adjusted R2 values of the CUMI outcome was the highest, followed by those of peanut particle size and eating impacts. (The descriptions have been revised in the ‘Result’ section, Page 12).

The statistically significant differences in the CUMI score between different tooth loss severity, as well as type and quality of dental prosthesis indicated that the discriminant validity of the CUMI in identifying different oral and denture status was better than those of eating impacts and masticatory performance. The highest adjusted R2 values of the CUMI models, followed by those of the masticatory performance and eating impacts, indicated that oral and denture status better explained the variances in the CUMI score compared with masticatory performance and eating impacts. The descriptions have been written in the ‘Discussion’ section, Page 16, 1st Paragraph.

Sincerely yours,

Wacharasak Tumrasvin

Corresponding author

---

## [Decision Letter · Decision Letter 1]

2 Aug 2021

PONE-D-21-02795R1

Masticatory Index for Patients Wearing Dental Prosthesis as Alternative to Conventional Masticatory Ability Measures

PLOS ONE

Dear Dr. tumrasvin,

Thank you for submitting your manuscript to PLOS ONE. After careful consideration, we feel that it has merit but does not fully meet PLOS ONE’s publication criteria as it currently stands. Therefore, we invite you to submit a revised version of the manuscript that addresses the points raised during the review process.

The reviewers and editor do not feel that the reviewer 1's comments were thoroughly addressed from the last revision. Please carefully address the comments and revise it before you resubmit.

We look forward to receiving your revised manuscript.

Kind regards,

Sompop Bencharit, DDS, MS, PhD, FACP

Academic Editor

PLOS ONE

Reviewers' comments:

Reviewer's Responses to Questions

**Comments to the Author**

1. If the authors have adequately addressed your comments raised in a previous round of review and you feel that this manuscript is now acceptable for publication, you may indicate that here to bypass the “Comments to the Author” section, enter your conflict of interest statement in the “Confidential to Editor” section, and submit your "Accept" recommendation.

Reviewer #1: (No Response)

Reviewer #2: (No Response)

2. Is the manuscript technically sound, and do the data support the conclusions?

Reviewer #1: Partly

Reviewer #2: Yes

3. Has the statistical analysis been performed appropriately and rigorously? 

Reviewer #1: No

Reviewer #2: I Don't Know

4. Have the authors made all data underlying the findings in their manuscript fully available?

Reviewer #1: No

Reviewer #2: Yes

5. Is the manuscript presented in an intelligible fashion and written in standard English?

Reviewer #1: Yes

Reviewer #2: Yes

6. Review Comments to the Author

Reviewer #1: I do not see any relevant change in the manuscript. Thus there is no reason to reconsider my recommendation

Reviewer #2: The study design using specific asian food makes the study hard to reproduce or repeat.

The previous comments were not responded reasonably especially comment 2, 4 and 5.

Is there any positive control in this study? please discuss.

7. PLOS authors have the option to publish the peer review history of their article (what does this mean?). If published, this will include your full peer review and any attached files.

Reviewer #1: No

Reviewer #2: No

---

## [Author Response · Author response to Decision Letter 1]

20 Aug 2021

The authors are pleased to submit our revised manuscript ID. PONE-D-21-02795R1, entitle ‘Masticatory Index for Patients Wearing Dental Prosthesis as Alternative to Conventional Masticatory Ability Measures’. The requested revisions have been made in the manuscript in track changes, and our point-by-point responses are below.

Reviewer #1: 

 The authors would like to apologize for our misunderstandings regarding the first revision of the manuscript. Therefore, we have added the descriptions in the second revised version of the manuscript according to your recommendations. 

1. Comment: The authors developed a food questionnaire based to the reports of only sixty individuals where 40 % were provided with complete dentures and RPDs and only 10% FPD or dentate. This may be already decisively biased as it is known that especially patients with CDs tend to select food that they can mince. 

Response: 

Initially, 40 patients (37.5% CD, 37.5% RPD, and 10% FPD and dentate) were asked about the typical food types they had eaten during the past week, and the food types they had difficulty in chewing or would like to eat but could not chew at all. Then, the additional 20 patients were interviewed. We found that the most regularly-consumed and rarely-consumed foods obtained from 40 and 60 patients were similar. Therefore, the Phase I comprised a total of 60 patients (40% CD, 40% RPD, 10% FPD and 10% dentate individuals). A higher proportion of removable dentures was included because they reported a greater variety of food types and textures compared with FPD and dentate individuals. The wide variety of food type was due to the variation in denture quality and oral status among the RPD and CD wearers. Then, 80 food types were obtained from the interview. The increased food varieties help improve discriminant validity of the masticatory index. These descriptions have been added in the ‘Materials and Methods’ section, ‘Phase I’ subsection (Page 6).

2. Comment: As the food items are special at least for Asian food the approach is very limited on worldwide perspective. On the other hand, peanuts that the authors use as a comparison are more or less available worldwide.

Response: The food items in the present questionnaire are special for Asian food, however, it is considered for worldwide use because the Asian-living culture and Asian populations are prevalent worldwide. In addition, the present study aimed not only to develop the questionnaire, but also to propose a concept of developing a questionnaire for masticatory ability evaluation in patients wearing different types of dental prosthesis. The authors believe that the readers can apply the concept and protocol of the present study to develop their own questionnaires, to be used in other cultures and countries. These descriptions have been added in the ‘Discussion’ section, 1st Paragraph, Page 19.

 Although multiple sieve method of peanut mastication is worldwide used, it consumes more time and resources to perform than the subjective measures. The whole evaluation process takes 2 days to obtain the result since collecting the comminuted peanut particles from patients, drying the comminuted peanut particles overnight, and then, analyzing the peanut particle size on the next day. Therefore, peanut mastication might be practical only in clinical study or research. These descriptions have been added in the ‘Discussion’ section, 1st Paragraph, Page 17.

3. Comment: The different chewing difficulty levels claimed for the food items were never assessed and /or verified.

Response: The chewing difficulty levels were obtained from participants’ response, and the food items were ranked from the highest to lowest masticatory score. In addition to the participants’ responses, three experts in prosthodontics scored the masticatory difficulty grade of each food item, and the values were compared with those obtained from all participants. Weighted Kappa scores ranging from 0.75–0.87 were calculated, indicating 90–95% agreement between the experts and participants. These descriptions have been added in the ‘Materials and Methods’ section, ‘Phase II - CUMI assessment’ subsection (Page 8).

The different chewing difficulty levels were verified between dental experts and patients. However, we did not use any specific instrument to assess masticatory difficulty level of each food item because it includes mixed properties of the food such as hardness, toughness, stickiness, slipperiness, and fibrousness. Therefore, no specific instrument can comprehensively determine these properties and verify the outcome. These descriptions have been written in the ‘Discussion’ section, 2nd Paragraph, Page 16.

4. Comment: The statement that the developed masticatory index reflects the patients’ ability better the conventional subjective or objective methods is not supported by the data.

Response: The developed masticatory index reflects the patients’ ability better than the conventional subjective and objective methods was supported by discriminant validity as shown by a dose-response relationship between the decreased CUMI score and increased tooth loss severity, and unacceptable denture quality. To support the statement, the revisions have been made in the ‘Abstract’ and ‘Discussion’ sections as follows: 

- In ‘Abstract section’, the revisions regarding discriminant validity have been made in the Methods, Results, and Conclusion.

- In ‘Result’ section’ (Page 14), the revision has been made in the multiple regression analysis of the CUMI stated that “After adjusting for age and sex, there was a significant dose-response relationship between an increased CUMI score, greater tooth loss severity, and unacceptable denture quality. Meanwhile, a dose-response relationship was not shown in the peanut particle size and eating impact models. For type of dental prosthesis model, the peanut particle size was significantly different only between FPD and CD, whereas the oral impact was different only between the acceptable and unacceptable denture quality. For both oral and denture status models, the adjusted R2 values of the CUMI outcome was the highest, followed by those of peanut particle size and eating impacts. Therefore, the CUMI demonstrated better discriminant validity than the peanut particle size and eating impact models.”

- In ‘Discussion’ section, 1st paragraph: “Since significant dose-response relationships were reported only between an increased CUMI score and greater tooth loss severity, and unacceptable denture quality, the CUMI demonstrated better discriminant validity than the conventional subjective and objective measures. The results indicated that the developed masticatory index better reflected patients’ masticatory ability compared with conventional subjective and objective measures.” 

Reviewer #2: 

1. Comments: The study design using specific Asian food makes the study hard to reproduce or repeat.

Response: The food items in the present questionnaire are special for Asian food, however, it is considered for worldwide use because the Asian-living culture and Asian populations are prevalent worldwide. In addition, the present study aimed not only to develop the questionnaire, but also to propose a concept of developing a questionnaire for masticatory ability evaluation in patients wearing different types of dental prosthesis. These descriptions have been added in the ‘Discussion’ section, 1st Paragraph, Page 19.

2. Comments: The previous comments were not responded reasonably especially comment 2, 4 and 5.

Response: The authors have responded to the previous comments especially 2, 4 and 5 and have revised them in the manuscript according to the reviewer’s recommendations.

3. Comments: Is there any positive control in this study? please discuss.

Response: In this study, the FPD group (dentate individuals who had at least 26 remaining natural teeth) served as a positive control because they showed the least frequent eating impact, and the highest masticatory performance and CUMI score. Approximately 79% of them could easily chew all food items or get a full CUMI score. The descriptions have been added in the ‘Discussion’ section (2nd Paragraph, Page 18) and mentioned in the “Materials and Methods” section.

Sincerely yours,

Wacharasak Tumrasvin

Corresponding author

---

## [Decision Letter · Decision Letter 2]

3 Jan 2022

PONE-D-21-02795R2Masticatory Index for Patients Wearing Dental Prosthesis as Alternative to Conventional Masticatory Ability MeasuresPLOS ONE

Dear Dr. tumrasvin,

Thank you for submitting your manuscript to PLOS ONE. After careful consideration, we feel that it has merit but does not fully meet PLOS ONE’s publication criteria as it currently stands. Therefore, we invite you to submit a revised version of the manuscript that addresses the points raised during the review process.

Please check the terms with the Glossary of Prosthodontics. Please see comments from the reviewer. Please submit your revised manuscript by Feb 17 2022 11:59PM. If you will need more time than this to complete your revisions, please reply to this message or contact the journal office at plosone@plos.org. Please include the following items when submitting your revised manuscript:A rebuttal letter that responds to each point raised by the academic editor and reviewer(s). You should upload this letter as a separate file labeled 'Response to Reviewers'.A marked-up copy of your manuscript that highlights changes made to the original version. You should upload this as a separate file labeled 'Revised Manuscript with Track Changes'.An unmarked version of your revised paper without tracked changes. You should upload this as a separate file labeled 'Manuscript'.If applicable, we recommend that you deposit your laboratory protocols in protocols.io to enhance the reproducibility of your results. Protocols.io assigns your protocol its own identifier (DOI) so that it can be cited independently in the future. For instructions see: https://journals.plos.org/plosone/s/submission-guidelines#loc-laboratory-protocols. Additionally, PLOS ONE offers an option for publishing peer-reviewed Lab Protocol articles, which describe protocols hosted on protocols.io. Read more information on sharing protocols at https://plos.org/protocols?utm_medium=editorial-email&utm_source=authorletters&utm_campaign=protocols.

We look forward to receiving your revised manuscript.

Kind regards,

Sompop Bencharit, DDS, MS, PhD, FACP

Academic Editor

PLOS ONE

Journal Requirements:

Reviewers' comments:

Reviewer's Responses to Questions

**Comments to the Author**

1. If the authors have adequately addressed your comments raised in a previous round of review and you feel that this manuscript is now acceptable for publication, you may indicate that here to bypass the “Comments to the Author” section, enter your conflict of interest statement in the “Confidential to Editor” section, and submit your "Accept" recommendation.

Reviewer #3: (No Response)

2. Is the manuscript technically sound, and do the data support the conclusions?

Reviewer #3: Yes

3. Has the statistical analysis been performed appropriately and rigorously? 

Reviewer #3: Yes

4. Have the authors made all data underlying the findings in their manuscript fully available?

Reviewer #3: Yes

5. Is the manuscript presented in an intelligible fashion and written in standard English?

Reviewer #3: Yes

6. Review Comments to the Author

Reviewer #3: The authors did a good job in addressing previous reviews and comments. the following are suggested modifications so that the manuscript can be acceptable for publication:

1. In the introduction, authors keep using the term "dental prosthesis" which is a little bit confusing as they cite articles that report on complete dentures. Please unify terminology through the introduction and manuscript.

2. Add a paragraph describing the limitations of this study and the drawn conclusions especially considering the subjects number.

7. PLOS authors have the option to publish the peer review history of their article (what does this mean?). If published, this will include your full peer review and any attached files.

Reviewer #3: No

---

## [Author Response · Author response to Decision Letter 2]

6 Jan 2022

The authors would like to give thanks for your time spent in peer-reviewing our manuscript. We are pleased to submit our revised manuscript ID. PONE-D-21-02795R2, entitle ‘Masticatory Index for Patients Wearing Dental Prosthesis as Alternative to Conventional Masticatory Ability Measures’. The requested revisions have been made in the manuscript in track changes, and our point-by-point responses are below.

Reviewer #3: 

1. Comment: The authors did a good job in addressing previous reviews and comments. The following are suggested modifications so that the manuscript can be acceptable for publication.

Response: Thank you for your feedback and comments.

2. Comment: In the introduction, authors keep using the term "dental prosthesis" which is a little bit confusing as they cite articles that report on complete dentures. Please unify terminology through the introduction and manuscript.

Response: A terminology of ‘dental prosthesis’ in this manuscript context has been defined in the 1st paragraph of ‘Introduction’ section; “In this context, a dental prosthesis refers to fixed partial denture (FPD), removable partial denture (RPD), and complete denture (CD).” The references have been revised throughout the manuscript to ensure that the cited articles cover the term ‘dental prosthesis’.

3. Comment: Add a paragraph describing the limitations of this study and the drawn conclusions especially considering the subjects number.

Response: The limitations of this study has been written in a separated paragraph. The limitation, considering the number of participants, has been added in the last paragraph of the ‘Discussion’ section, and in the ‘Conclusion’ section.

Sincerely yours,

Wacharasak Tumrasvin

Corresponding author on behalf of all authors

---

## [Editor Report · Decision Letter 3]

12 Jan 2022

Masticatory Index for Patients Wearing Dental Prosthesis as Alternative to Conventional Masticatory Ability Measures

PONE-D-21-02795R3

Dear Dr. tumrasvin,

We’re pleased to inform you that your manuscript has been judged scientifically suitable for publication and will be formally accepted for publication once it meets all outstanding technical requirements.

Kind regards,

Sompop Bencharit, DDS, MS, PhD, FACP

Academic Editor

PLOS ONE

Additional Editor Comments (optional):

Thank you for the revision and responses to the reviewers.
---

## [Editor Report · Acceptance letter]

17 Jan 2022

PONE-D-21-02795R3 

Masticatory Index for Patients Wearing Dental Prosthesis as Alternative to Conventional Masticatory Ability Measures 

Dear Dr. Tumrasvin:

I'm pleased to inform you that your manuscript has been deemed suitable for publication in PLOS ONE. Congratulations! Your manuscript is now with our production department. 

Kind regards, 

on behalf of

Dr. Sompop Bencharit 

Academic Editor

PLOS ONE